# Global Dimming and Brightening Features during the First Decade of the 21st Century

**Nikolaos Hatzianastassiou [1,\*], Eleftherios Ioannidis [1,2], Marios-Bruno Korras-Carraca [1,3], Maria Gavrouzou [1], Christos D. Papadimas [1], Christos Matsoukas [3], Nikolaos Benas [4], Angeliki Fotiadi [5], Martin Wild [6] and Ilias Vardavas [7]**

[1]  Laboratory of Meteorology, Department of Physics, University of Ioannina, 45110 Ioannina, Greece; elefth.ioannidis@gmail.com (E.I.); koras@env.aegean.gr (M.-B.K.-C.); gavrouzou.m@gmail.com (M.G.); chpapadimas@gmail.com (C.D.P.)

[2]  LATMOS/IPSL, Sorbonne Université, UVSQ, CNRS, 75252 Paris, France

[3]  Department of Environment, University of the Aegean, 81100 Mytilene, Greece; matsoukas@aegean.gr

[4]  R & D Satellite Observations Department, Royal Netherlands Meteorological Institute (KNMI), 3730 AE De Bilt, The Netherlands; benas@knmi.nl

[5]  Department of Environmental Engineering, University of Patras, 74148 Agrinio, Greece; afotiadi@upatras.gr

[6]  Institute for Atmospheric and Climate Science, ETH Zurich, 8092 Zurich, Switzerland; martin.wild@env.ethz.ch

[7]  Department of Physics, University of Crete, 71003 Heraklion, Crete, Greece; vardavas@uoc.gr

\*  Correspondence: nhatzian@uoi.gr

**Abstract:** Downward surface solar radiation (SSR) trends for the first decade of the 2000s were computed using a radiative transfer model and satellite and reanalysis input data and were validated against measurements from the reference global station networks Global Energy Balance Archive (GEBA) and Baseline Surface Radiation Network (BSRN). Under all-sky conditions, in spite of a somewhat patchy structure of global dimming and brightening (GDB), an overall dimming was found that is weaker in the Northern than in the Southern Hemisphere ($-2.2$ and $-3.1$ W m$^{-2}$, respectively, over the 2001–2009 period). Dimming is observed over both land and ocean in the two hemispheres, but it is more remarkable over land areas of the Southern Hemisphere. The post-2000 dimming is found to have been primarily caused by clouds, and secondarily by aerosols, with total cloud cover contributing $-1.4$ W m$^{-2}$ and aerosol optical thickness $-0.7$ W m$^{-2}$ to the global average dimming of $-2.65$ W m$^{-2}$. The evaluation of the model-computed GDB against BSRN and GEBA measurements indicates a good agreement, with the same trends for 65% and 64% of the examined stations, respectively. The obtained model results are in line with other studies for specific world regions and confirm the occurrence of an overall solar dimming over the globe during the first decade of 21st century. This post-2000 dimming has succeeded the global brightening observed in the 1990s and points to possible impacts on the ongoing global warming and climate change.

**Keywords:** surface solar radiation; climate; dimming; brightening; model; stations; satellite data

## 1. Introduction

The downward surface solar radiation (SSR), apart from long-term variations, has been shown to have undergone decadal variations that are documented on a regional and global scale since the middle of the 20th century. Land-based observations, modeling studies, and satellite-derived records, point to widespread decreases in SSR from the 1950s to 1980s ("global dimming") followed by a partial recovery and increase ("brightening") during the late 1980s and 1990s [1,2]. Such decadal scale changes in SSR, also known as "global dimming and brightening" or "GDB", play an important role in various aspects

of climate change, namely in the modulation of greenhouse gas-induced warming, the hydrological cycle, the cryosphere or the terrestrial carbon cycle, and plant growth [3]. Therefore, the monitoring of decadal scale changes of GDB is very important, especially in the context of the ongoing global warming and climate changes.

To date, a complete assessment of GDB beyond 2000 and during the first decade of 21st century does not exist. Updates of the phenomenon beyond 2000 either partly cover the first decade or focus on specific world regions [4–18], namely the United States, Canada, Israel, Europe, India, China, New Zealand, while most of them are based on surface measurements. The existing updates suggest mixed tendencies. [1], using SSR measurements from global surface networks such as GEBA (Global Energy Balance Archive), WRDC (World Radiation Data Center) and BSRN (Baseline Surface Radiation Network) for the period 2000–2005 found a less distinct brightening after 2000 than in the 1990s, arising from brightening in the USA and Europe, a level-off at Japanese sites, and a renewed dimming in India, China, Iran, and in the Gulf region. However, given the large scale of GDB, reliable conclusions cannot be drawn based only on surface measurements. This is only possible using satellite-derived SSR fluxes that provide global coverage. Very few such attempts were made to extend GDB beyond 2000 based on satellite-derived SSR, indicating tendencies towards a renewed dimming in the first years after 2000. Some of them provide long temporal coverage, but not a global one, for example [16], for Europe, until 2015, [17] for the eastern Mediterranean until 2013, while others ensure global coverage, but do not cover the entire decade, for example [19], until 2004. In the work by [20] based on radiative transfer computations and information from satellites, reanalyses and surface measurements, an inter-hemispherical asymmetry of GDB, with a just slight (0.17 W m$^{-2}$) brightening in the Northern Hemisphere and a clear dimming ($-2.88$ W m$^{-2}$) in the Southern Hemisphere during 2001–2006, has been reported.

Here, an update on GDB for the first decade of 2000s is attempted, extending the results of [20]. The results of the present study cover the entire globe, and span the period 2000–2009, which is marked by the end of the International Satellite Cloud Climatology Project (ISCCP-D2) dataset. ISCCP-D2 has been the longest reliable global climatological dataset providing cloud optical properties that enable radiative transfer model computations, and it has been extensively used worldwide [21] also constituting a basis for the latest subsequent Intergovernmental Panel on Climate Change (IPCC) reports [22–24]. The results of the present study are obtained using a reliable detailed spectral radiative transfer model and input data from global satellite (ISCCP, MODIS (Moderate Resolution Imaging Spectroradiometer) C006) and NCEP/NCAR (National Centers for Environmental Predictions and National Center for Atmospheric Research) reanalysis datasets, with the aim of providing local, regional, and hemispherical aspects of the post-2000 GDB phenomenon. In the present study, MODIS C006 aerosol data are used instead of MODIS C005 ones used by [20], which allows us to incorporate desert areas that were missing in the previous study of 2012. Moreover, the present study emphasizes the identification of the causes of post-2000 GDB and its qualitative and quantitative attribution to physical parameters that are drivers of SSR. The quality of the model estimates of SSR is ensured through comparisons against quality surface station measurements from the reference GEBA and BSRN global networks, while they are also compared with other published works that mainly focus on specific world regions and rely on surface measurements.

## 2. Materials and Methods

### 2.1. Radiation Transfer Model

The radiative transfer model (RTM) used here to compute the SSR fluxes is the same as the one used in the study by [20], allowing direct intercomparisons. For a complete and in-depth description of the model, the reader is also referred to [25]. The model solves the radiative transfer equation for an absorbing and multiple-scattering atmosphere separately in 118 wavelengths in the ultraviolet, visible, and part of the near-IR range (0.2–1.0 μm), and in 10 spectral bands in the near- and mid-IR range

(1.0–10 µm), using the modified delta-Eddington method of [26]. It takes into account processes that affect radiation transfer, including Rayleigh scattering and atmospheric absorption in the UV and in the visible ($O_3$ Hartley-Huggins and Chappuis bands), as well as in the near-IR by $H_2O$, $CO_2$, and $CH_4$. It also includes scattering and absorption by aerosols and clouds, considering the latter in two phases (liquid and ice) and three atmospheric layers (low, middle, and high). Lambertian reflectance is assumed at the Earth's surface over land, while over ocean Fresnel reflection is used, also taking into account imperfections in the smoothness of the sea surface [25].

*2.2. Model Input and Validation Data*

The RTM has been used in the past [27–29] in various temporal (instantaneous, daily, monthly) and spatial scales (ranging from a few kilometers to 2.5° latitude-longitude). The present model configuration is the same as in [20], namely using monthly mean input data at 2.5° spatial resolution. Input data are also mostly as in [20], except for aerosol optical thickness (AOT), which is taken from the MODIS-Terra Collection 006 database here instead of MODIS C005 in the previous work. MODIS C006 data include the use of the Deep-Blue algorithm [30,31], which retrieves aerosol properties over both vegetated and arid areas. MODIS is an advanced spectroradiometer operating on board NASA's Terra and Aqua satellites, observing the Earth from sun-synchronous morning (Terra) and afternoon (Aqua) orbits. MODIS C006 AOT is a widely used and validated product [32]. Here, the level-3 monthly version was used. MODIS AOT was supplemented by aerosol asymmetry parameter and single scattering albedo (SSA) from the Global Aerosol Data Set (GADS) climatology [33]. All cloud properties required by the model, namely cloud amount, phase, optical depth, cloud-top properties (pressure and temperature), and geometrical thickness, at three cloud layers, are included in the ISCCP-D2 data set. ISCCP is among the most comprehensive global cloud properties data sets, combining observations from both geostationary and polar orbiters, with data availability extending back in the early 1980s [34]. Atmospheric temperature and humidity profiles, as well as water vapor data, also required for the radiative transfer calculations, were taken from the data set provided by NCEP/NCAR. This data set is based on a reanalysis system that assimilates quality-controlled data from various sources, ranging from ground measurements to satellite retrievals [35]. Regarding atmospheric constituents, $O_3$ is also available in ISCCP-D2, originally retrieved from the TIROS (Television Infrared Observation Satellite) Operational Vertical Sounder (TOVS) sensor; based on [36], the mixing ratio of $CH_4$ used in the model is set equal to 1.774 ppmv; $CO_2$ has also been updated compared to [20], and set equal to 375 ppmv, with the new value being more representative of the period examined. The spatial (2.5° latitude-longitude) and temporal (monthly) resolution is constrained by the ISCCP-D2 data and other data sets were re-gridded to the 2.5° resolution and temporally averaged to produce monthly values.

The temporal coverage of model SSR fluxes, constrained by the availability of all RTM input data, is from March 2000 to December 2009. However, SSR changes and tendencies were computed for the period from January 2001 to December 2009, in order to ensure complete annual coverage by SSR fluxes, which is essential for adequate computations of tendencies and trends. As in [20], the RTM SSR fluxes were validated again against the GEBA database [37–40] and the BSRN database [26,29] of the World Climate Research Program (WCRP) Global Climate Observing System (GCOS). The GEBA is a database for the worldwide measured energy fluxes at the Earth's surface maintained at ETH Zurich [41]. It currently contains 2500 stations with 500,000 monthly mean values of various surface energy balance components. By far the most widely measured quantity is the downward solar radiation at the Earth's surface, predominantly measured by pyranometers. Many of these records cover several decades, extending back to the 1950s, 1960s, or 1970s. GEBA compiles monthly data from a variety of sources, namely from the World Radiation Data Centre (WRDC) in St. Petersburg, from National Weather Services, from different research networks such as BSRN, from peer-reviewed publications, project and data reports, as well as from personal communications. Quality checks are applied to test for gross errors in the dataset. The relative random error (root mean square error/mean) of the downward solar radiation values is at 5% for the monthly means and 2% for the yearly means [38]. The

BSRN [39,42] is a worldwide network of radiation sites measuring at the highest possible accuracy with well-calibrated instruments and known accuracy. BSRN became operational in the early 1990s with a few sites and has gradually been growing to contain now more than 60 sites in various climate zones, which report their data to the BSRN Archive at the Alfred Wegener Institute (AWI) [43]. The BSRN data are recorded and stored at high temporal resolution (minute data). BSRN sites are further requested to record shortwave radiation not only in terms of its total flux (measured with a pyranometer), but also separately in terms of the direct shortwave flux (measured with a pyrheliometer) and the diffuse shortwave flux (measured with a shaded pyranometer). The accuracy of the BSRN solar and thermal radiation measurements is discussed in [44]. From the total number of more than 1600 GEBA stations, 105 GEBA stations having sufficiently complete records, i.e., for at least 81 out of the total 108 months from January 2001 to December 2009, were used. As to the even higher accuracy BSRN fluxes, from the few dozens BSRN sites dating back to 1992, 20 stations were chosen, having again at least 81 out of 108 monthly records available from January 2001 to December 2009.

### 2.3. Methodology

The inter-annual variability of model SSR fluxes, either on a $2.5° \times 2.5°$ latitude–longitude cell or larger scale (hemispherical) basis, as well as of station (GEBA and BSRN) SSR fluxes, is examined using the deseasonalized anomalies of monthly SSR values, which were computed by subtracting from each monthly SSR value the corresponding long-term (2001–2009) averaged value. This process minimizes the seasonal component of the variability, which is assumed to be periodic with constant amplitudes and thus does not have much impact on the long-term variability. Given that the change (tendency) in SSR is not a simple linear change, it is quantified by applying linear regression on the time series of deseasonalized SSR anomalies. More specifically, we apply a simple linear trend model of the form:

$$Y = \mu + \omega{\cdot}x$$

where $Y$ is the SSR deseasonalized anomalies, $x$ is time (months), $\mu$ is the intercept (the $Y$ value when $x = 0$), and $\omega$ is the magnitude of the gradient (slope) of the trend line (change of SSR per month). The SSR changes ($\Delta$SSR) during 2001–2009 were computed by multiplying the values of the SSR slopes (obtained from the linear regression) by the temporal extent of our study (108 months). The statistical significance of the SSR tendencies was assessed at the 95% confidence level by applying the non-parametric Man-Kendall test [45,46] to the time series of monthly SSR anomalies. The Mann-Kendall is a commonly used statistical test that is frequently used to quantify the significance of trends in time series of atmospheric parameters [47,48].

In the analysis, in order to investigate the causality of GDB and the relationship between the estimated GDB ($\Delta$SSR) and the changes of single (every) GDB driver, for example, cloud amount and aerosol optical thickness, bivariate correlation analysis is applied and the values of correlation coefficients (R) are computed. The R values are also computed when assessing the performance of the RTM against surface GEBA and BSRN measurements, i.e., comparing their SSR fluxes. In both cases, Pearson correlation coefficients, measuring the strength of linear association between the paired variables, are computed using linear regression.

The RTM SSR and $\Delta$SSR fluxes are originally computed on a $2.5° \times 2.5°$ latitude–longitude cell and monthly basis. Subsequently, when averaging spatially and temporally, strict availability criteria are applied. More specifically, the severe criterion of availability of 100% of all RTM input data and computed SSR fluxes in their time series is applied, which is essential for ensuring robust statistical results. Moreover, for the computation of annual mean values, either on individual or 9-year basis, the availability of all months and years is required. Finally, the computation of hemispherical averages takes into account the different spatial extent (areal coverage) of geographical cells found in different latitudinal zones, associated with the Earth's non-sphericity.

## 3. Results

### 3.1. Features of Global Dimming and Brightening

Figure 1 displays the model computed tendencies of deseasonalized anomalies of SSR for the period 2001–2009 along with their statistical significance, for each $2.5° \times 2.5°$ cell. Thus, the presented results allow us to examine the existence, the magnitude, and the significance of the detected SSR tendencies by the model. Solar brightening (increasing SSR, yellowish and reddish colors) and dimming (decreasing SSR, bluish colors) patterns, being statistically significant to a considerable extent, appear over extended world regions, which are sometimes mixed, occurring in adjacent world areas. Thus, solar dimming and brightening appear in the northern and southern USA, as well as in the northern and southern parts of South America, respectively. Similarly, opposite SSR tendencies are also seen in Africa, Asia, as well as over the Atlantic and Pacific Oceans, although there is an apparent predominance of dimming, i.e., declining SSR fluxes. On the other hand, there are world areas, like Europe and the Mediterranean, where there is a clear indication of brightening. Overall, dimming slightly dominates brightening, appearing in 54% of the geographical cells with computed SSR tendencies (versus 46% of cells with brightening). The obtained results show that the tendencies are statistically significant (at the 95% confidence level) over a considerable part of the globe, (dark points in Figure 1), constituting 28% of the overall number of cells with computed SSR tendencies. In general, the stronger dimming and brightening tendencies (deeper blue and red colors, respectively) observed in Figure 1 are significant. This is the case for the dimming over the Arabian Peninsula, South America, Australia, and China, as well as the brightening over the western Pacific Ocean, the southern Indian Ocean or southern Europe. On the other hand, there are other areas, namely North America and Africa, where the SSR tendencies are not found to be statistically significant. In total, 63% of cells with statistically significant tendencies have positive tendencies and 37% have negative ones. Thus, it appears that brightening is dominating when areas with statistically significant GDB results are considered, while dimming is predominant in general, i.e., when all results (all world areas with GDB results) are considered.

The GDB regime during 2001–2009 (Figure 1) is similar to some extent with the corresponding one presented in Figure 1 of the study by [20], obtained with the same RTM and similar data, but using MODIS C005 data instead of C006 (see Section 2) and covering the shorter period 2001–2006. It should be noted that Figure 1 here has a more limited geographical coverage than the corresponding one of [20], because of the application of the severe criterion of availability of 100% of all RTM input data and computed SSR fluxes. This resulted in a loss of ΔSSR for grids to the north of 50° N and to the south of 45° S because of the incomplete time series of SSR fluxes, mainly due to missing MODIS AOT. Although the spatial coverage of the GDB results here is decreased with respect to the previous study (where the coverage reaches 75° N and 65° S), this is preferred in order to ensure statistical significance and robustness. On the other hand, the present study ensures geographical coverage over highly reflecting global deserts, namely the Sahara, Arabian Peninsula, and Asian deserts, which were missed in [20]. This is achieved here due to the MODIS C006 data, which include the use of the Deep-Blue algorithm, instead of MODIS C005 data used in [20].

The obtained GDB results for the first decade of 2000 (Figure 1) extend the corresponding patterns for the period 2001–2006 [20], either confirming or changing the labeling (dimming or brightening) of GDB. Thus, there are areas over which the reported dimming/brightening by [20] is confirmed in Figure 1. This is the case, for example, of solar brightening in Europe, which, however, is slightly more extended to the south, encompassing the northern coastline of Africa and the adjacent southernmost Mediterranean Sea areas, which were rather characterized by dimming in [20]. Otherwise, a widespread brightening is observed over Europe and the Mediterranean, in agreement with several surface or satellite-based studies [1,13,49–51]. This is also in agreement with the increasing tendency of SSR over Europe during the first decade of the 2000s reported by [16] based on two satellite-based records of EUMETSAT (European Organisation for the Exploitation of Meteorological Satellites) Satellite Application on Climate Monitoring (CM SAF). Other cases of agreement between the present study

and [20] are the brightening observed along the USA coastal areas of the Gulf of Mexico, the brightening over Argentina, and the dimming over central America, Brazil (Amazonian basin), Australia, and the southwestern Indian Ocean off the eastern coasts of southern Africa (around Madagascar). On the other hand, differences are also encountered between the GDB patterns in Figure 1 and the corresponding results of [20] for the 2001–2006 period (their Figure 1). Specifically, in the present study, there is no clear GDB pattern during 2001–2009 over North America, but there is contrasting dimming and brightening patterns over its northern and southern parts, respectively. On the contrary, a clear brightening pattern was reported by [20] for the same area and for the period 2001–2006. A different GDB pattern is also found over China, whereas clear dimming is found in Figure 1, opposite to the brightening over the largest part of China reported by [20]. It seems that the extension of the studied period to 2009, changed the identified GDB pattern in this world region. The results of Figure 1 are in agreement with the results of [34] who, using satellite data as well as station data from the GEBA network and the Meteorology agency of China, found dimming in China for the period 2001–2007. Another case of different GDB between this study and that of [20] is Italy. The brightening observed over the Italian peninsula in Figure 1, which is in agreement with the brightening reported by [14] for the same world area, estimated using SSR measurements from 54 stations distributed throughout Italy, is opposed to the dimming over the southern Italian peninsula during 2001–2006 reported by [20]. This underlines the importance of ensuring completeness in assessments of decadal scale tendencies of surface solar radiation. Finally, it should be noted that the present study provides information about GDB over the world deserts, which were absent in the study by [20]. It is found that dimming took place during 2001–2009 over the Sahara, Arabian Peninsula, and central Asian deserts, against a brightening observed over the Taklamakan and Gobi Asian deserts. The Sahara and Arabian Peninsula dimming is in line with the study from [52] and can be partially attributed to a decrease in precipitation leading to higher dust loads. Overall, a more widespread dimming is observed for the period 2001–2009, compared to 2001–2006 [20], which is quite clearer over the Pacific and Atlantic Oceans, as well as over continental areas in Asia, Africa, and America. The fact that adding 3 years can result in somewhat different tendencies is not strange, since adding/removing a few years, especially when relatively short periods are examined and tendencies are not statistically significant, can modify the results. Hence, the results of the present study can be taken as more definite regarding GDB during the first decade of the 2000s.

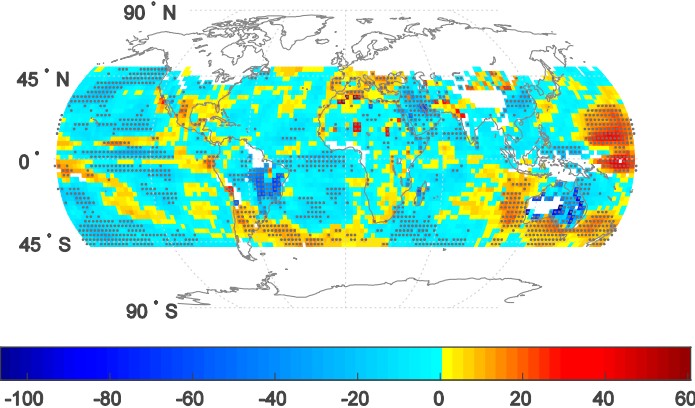

**Figure 1.** Tendencies of model-computed downward surface solar radiation anomalies (changes in surface solar radiation (SSR), in W m$^{-2}$) during the period 2001–2009. Dots indicate geographical cells for which the SSR tendencies are statistically significant.

Overall assessments of GDB on a hemispherical scale are useful since they can be related to global scale climate changes. Therefore, the anomalies of the hemispherical averages of SSR fluxes were computed, and their time series along with the corresponding time series of land and ocean areas of each hemisphere for the period 2001–2009 are overplotted in Figure 2a,b for the Northern and

Southern Hemispheres, respectively. The same information is given in Figure 3 of the study by [20] but for the period 2001–2006. Apart from the different temporal coverage, another difference is that, in the present study, the hemispherical averages refer to world areas up to 50° N and 45° S, whereas in [20] they would refer to areas reaching 70° N and 60° S. Although dimming is predominant in both the present study and the previous one, there are two major differences. The first difference concerns the Northern Hemisphere (NH), over which SSR has undergone a similar decrease over both land and oceans from 2001 to 2009, equal to −2.6 W m$^{-2}$ (or −0.29 W m$^{-2}$/year) and −2.2 W m$^{-2}$ (or −0.24 W m$^{-2}$/year) respectively, yielding an overall dimming of −2.2 W m$^{-2}$ (or −0.24 W m$^{-2}$/year). On the contrary, in [20], in the NH, SSR was found to have undergone a slight increase (brightening) of 0.17 W m$^{-2}$ (or 0.028 W m$^{-2}$/year) arising from a brightening of 0.44 W m$^{-2}$ (or 0.07 W m$^{-2}$/year) over land and a dimming of −0.75 W m$^{-2}$ (or −0.125 W m$^{-2}$/year) over ocean areas of NH. This difference may be attributed either to the different time periods or the different geographical coverage. In order to examine this, the tendencies of SSR in the present study were recomputed but for the period in [20], i.e., for 2001–2006. The results (Figure S2) yielded a dimming in both land and ocean areas of NH, equal to −2.2 W m$^{-2}$ (or −0.37 W m$^{-2}$/year) and −0.99 W m$^{-2}$ (or −0.165 W m$^{-2}$/year) respectively, resulting in an overall dimming of −1.1 W m$^{-2}$ (or −0.183 W m$^{-2}$/year). The strong similarity of these results (2001–2006) with those of Figure 2a of the present study (2001–2009), and the remaining differences with the results of Figure 3a of [20] for the same period, lead to the conclusion that the different geographical coverage is the most important reason for the difference between the Figure 2a of the present study and Figure 3a of [20]. The same analysis for SH (South Hemisphere), revealed that the time series of SSR anomalies of Figure 2 but for the period 2001–2006, yields a dimming of −8.5 W m$^{-2}$ (or −1.42 W m$^{-2}$/year) over land, a dimming of −1.56W m$^{-2}$ (or −0.26W m$^{-2}$/year) over ocean, and an overall dimming of −1.88W m$^{-2}$ (or −0.31 W m$^{-2}$/year) over land and ocean areas of SH. These results agree in terms of sign with both the results of Figure 2b of the present study and Figure 3b of the study by [20]. However, in terms of the magnitude of GDB, they appear to disagree with the results of Figure 2a. Specifically, although the dimming in SH land remains stronger than in SH ocean, the degree of difference is smaller (−8.5 W m$^{-2}$ versus −1.51 W m$^{-2}$ for 2001–2006, against −14.2 W m$^{-2}$ versus −1.4 W m$^{-2}$ for 2001–2009). Namely, it seems that the already occurring dimming over SH land areas until 2006 was further intensified afterwards, i.e., through to 2009. At the same time, the ratio of SH land/ocean dimming in this study, both for 2001–2006 (5.6) and 2001–2009 (10.1) periods, is opposite to that shown in the study of [20] for 2001–2006, where it was stronger over ocean than land areas (0.37). This finding underlines the importance of both the geographical and temporal coverage in assessing GDB at hemispherical global and decadal timescales.

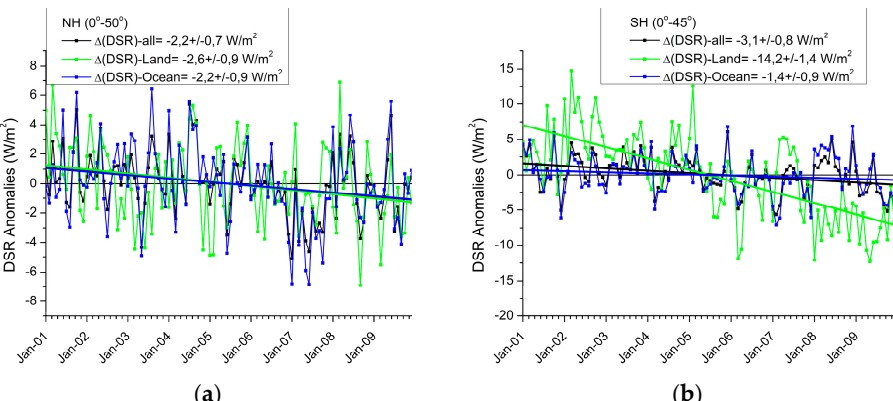

**Figure 2.** Time series of deseasonalized anomalies of monthly SSR fluxes averaged over land (green lines), ocean (blue lines), and land + ocean (black lines) regions of the Northern Hemisphere (**a**) and the Southern Hemisphere (**b**), over the period 2001–2009. The global dimming and brightening (GDB) magnitudes (SSR changes (ΔSSR), in W m$^{-2}$) and the associated standard error values over the period 2001–2009 are also given for each hemisphere.

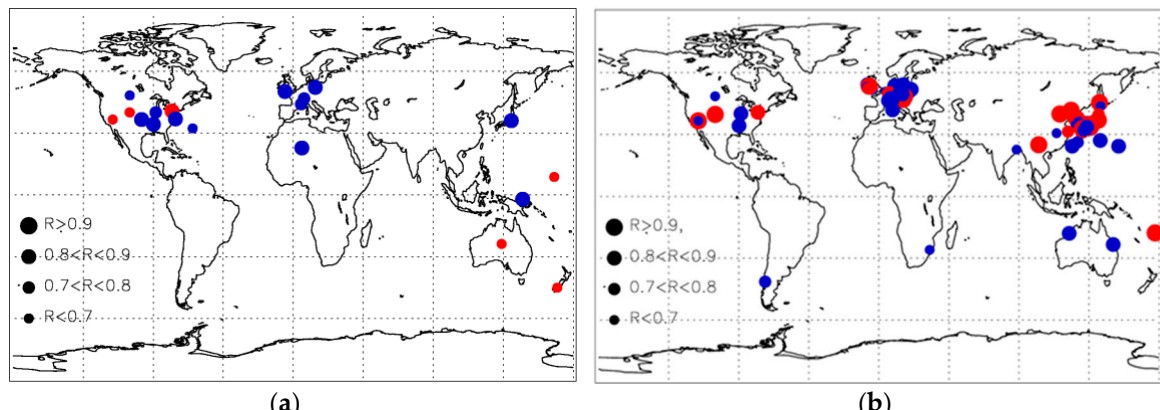

**Figure 3.** Comparison between model-computed tendencies of SSR anomalies (ΔSSR) during 2001–2009 and similar anomalies of the Baseline Surface Radiation Network (BSRN) (**a**) and the Global Energy Balance Archive (GEBA) (**b**) station measurements. Blue and red circles indicate BSRN and GEBA stations for which the sign of tendencies of model-computed and station-measured SSR anomalies (ΔSSR) agree and disagree, respectively. The size of circles indicates the magnitude of computed correlation coefficients R between model-computed and ground-measured SSR anomalies.

### 3.2. Evaluation of the RTM-Computed Global Dimming and Brightening

As done in [20], the model-computed GDB over 2001–2009 has been evaluated against available information from stations from the reference global networks of GEBA and BSRN. Here, the number of stations (fulfilling the availability criteria described in Section 2) is increased compared to the study by [20], from 91 to 105 for GEBA, and from 14 to 20 for BSRN. For each station, the tendencies of SSR from the GEBA/BSRN time series of monthly SSR anomalies were computed and compared to the corresponding tendencies of the time series of model SSR anomalies for the 2.5-deg geographical cells that include each GEBA or BSRN station. It should be noted that in the corresponding comparison made in [20] (their Figure 2), the tendencies of SSR and not of the SSR anomalies were computed and compared. Here, the use of SSR anomalies was preferred to the use of SSR fluxes themselves, for which a large R value would be dominated by the strong seasonal cycle of SSR. The model and station SSR tendencies (Figure 3a,b) are in quite good agreement, much like the work by [20]. The same tendencies of SSR (anomalies) are found in 67 (or 64%) out of 105 GEBA stations, against 54 out of 91 (or 60%) in the work by [20]. For BSRN stations, same tendencies are found here for 13 out of 20 (65%) BSRN stations, while the corresponding number in the work by [20] was 13 out of 14 stations. The differences between the present study and those in [20] should be attributed to the different time periods and the use of SSR anomalies instead of fluxes. The good agreement between the RTM and GEBA/BSRN GDB is also confirmed by the fact that the computed correlation coefficients (R) are higher than 0.8 for 43 (41%) GEBA stations and 9 (45%) BSRN stations, indicating a similar good performance of the model against both station networks. Our results (Figure S1) also confirm the ability of the model to match the variability of the GEBA and BSRN station SSR time series. More specifically, 16 out of 20 stations (80%) have differences in model and station 95% CI (confidence interval) values that are smaller than 30%, 12 stations (60%) have differences smaller than 20%, and seven stations (35%) have differences smaller than 10%. Even better statistics are found for the GEBA stations, for which 99 out of 105 (i.e., in 94% of stations) have differences smaller than 30%, 82 stations (78%) have differences smaller than 20%, and 57 stations (54%) have differences smaller than 10% (from which 33 have differences smaller than 5%).

### 3.3. Causes of Global Dimming and Brightening

Apart from the identification of the features of GDB during the first decade of 2000, the underlying physical reasons behind it have also been investigated. Clouds and aerosols are the main drivers of surface solar radiation, and both have been reported to be responsible for detected patterns of GDB in

the period before 2000. In order to examine their role for the post-2000 GDB assessed in this study, the tendencies of total cloud cover and aerosol optical thickness during 2001–2009 were computed at a 2.5° grid-level (similarly to the analysis for SSR, namely by applying linear regression to the time series of the aforementioned parameters' monthly anomalies) and are presented in Figure 4a,b, respectively. It should be noted that a similar analysis has been also done for other parameters, e.g., cloud optical thickness or incoming solar radiation at the top-of-atmosphere, but it is not given here since total cloud cover (TCC) and AOT were found to be the main drivers of 2001–2009 GDB. According to Figure 4a, there has been an absolute increase in TCC, on average global scale over 2001–2009, by 0.4%. This slight increase arises from mixed patterns of TCC changes all over the globe, with decreasing (bluish areas) and increasing tendencies (yellowish and reddish areas) each accounting for 50% of total grids with computed TCC tendencies. Note that the TCC tendencies in Figure 4a were found to be statistically significant (dark dots), according to the applied Mann-Kendall test, in 20% of the total number of cells with computed results. In 59% of them, there is a negative statistically significant tendency, whereas in 41%, there is a positive significant tendency. The comparison of Figure 4a (ΔTCC) with Figure 1 (ΔSSR) reveals an apparent inter-relationship between changes in total cloud cover and downward surface radiation, consisting of decreasing/increasing TCC and increasing/decreasing SSR. Thus, GDB patterns in Europe/the Mediterranean, North America, Pacific, and Indian oceans and South China Sea are in line with the corresponding patterns (increases and decreases) of total cloud cover over the same regions. More specifically, it is interesting to note that the decreasing TCC over the western Pacific (Figure 4a), which looks very similar to ENSO (El Niño–Southern Oscillation) patterns, is in line with increasing SSR over the same Pacific Ocean areas in Figure 1. Indeed, our study period (2001–2009) started with a noticeable La-Niña and was predominated and ended with an El-Niño, which explains both the TCC and SSR tendencies. However, clouds alone cannot explain the GDB patterns. For example, TCC is found to have decreased over India, while SSR is found to have decreased there, contrary to expectations based solely on TCC findings. The 2001–2009 changes of AOT (Figure 4b) indicate predominantly increasing tendencies over much of the globe (yellowish-reddish colored cells), yielding an average overall increase of 6.8% globally. More specifically, increased AOT is observed in 79.5% of the cells with computed AOT tendencies. The statistical significance of the tendencies of AOT in Figure 4b was examined by applying the Mann-Kendall test and the cells with statistically significant tendencies are marked by dark dots in Figure 4b. The AOT tendencies are significant in 26% of the total number of cells with computed results, namely over southern Europe, the Arabian Peninsula, China, and south Asia, broad areas of the Pacific Ocean, and the Arabian Sea. There is a prevalence of positive tendencies in them (59% versus 41% for positive and negative tendencies, respectively). Comparing Figure 4b (ΔAOT) with Figure 1 (ΔSSR), there are also some interconnections between the changes in AOT and SSR. For example, the increase in SSR over Europe (brightening) is in line with the corresponding decrease in AOT. Besides, the increase of SSR in the USA and Mexico areas surrounding the Gulf of Mexico is consistent with the decrease in AOT over those areas, while the decrease in SSR in Arabia and eastern China is interrelated with the increase of AOT there. However, similar to clouds, AOT alone cannot explain all the GDB patterns. Hence, other atmospheric parameters, such as cloud optical thickness and precipitable water, were considered in order to explain the GDB patterns.

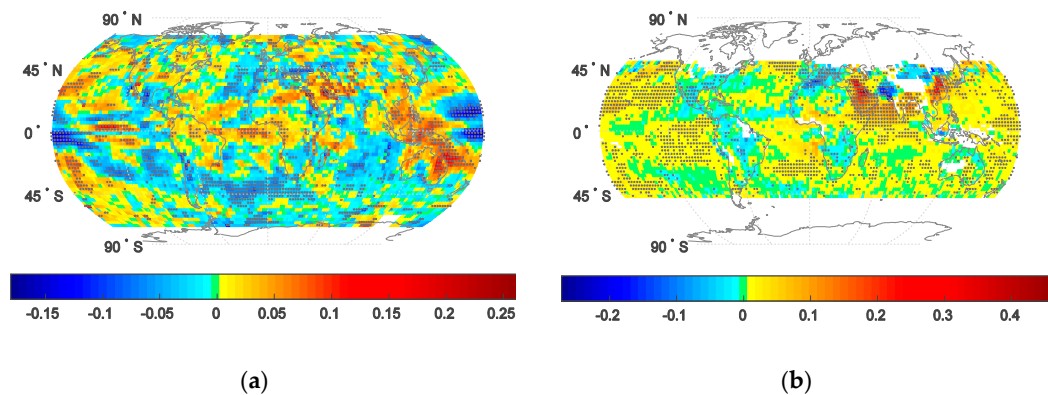

**Figure 4.** Tendencies of total cloud cover (**a**) and aerosol optical thickness (**b**) over the period 2001–2009. Total cloud cover (TCC) values are expressed in percentages (%) when multiplied by 100 and aerosol optical thickness (AOT) values are unitless. Dots indicate geographical cells for which the TCC and AOT tendencies are statistically significant.

The main parameters, V, that are drivers of SSR, have been correlated with ΔSSR, i.e., GDB, and the relevant scatter plots between ΔV, i.e., the changes of every parameter, and ΔSSR were produced (not shown here). Table 1 lists the computed correlation coefficients (R), and the agreement (percentage of the total number of matched data pairs lying in the 2nd and 4th quadrants) and disagreement (the percentage of points in the 1st and 3rd quadrants) between the tendencies of the most significant parameters and SSR. According to the computed R values it seems that total cloud cover played the most significant role for GDB (R = −0.58), while the high-level cloud optical thickness (especially absorption) has been the second most important factor of GDB (R = −0.47). On the other hand, aerosols, and more specifically AOT, have played a less significant role, yielding a correlation coefficient equal to −0.25. From the results of Table 1, it is clear that no single parameter alone can explain GDB, while all parameters contribute to a varying degree. Therefore, in order to determine the contribution of each parameter to GDB, we run the RTM repeatedly, keeping every single parameter "frozen" at its initial conditions in 2001, i.e., in the first year of the study period. Subsequently, from the difference between the main RTM run, the one having all parameters activated, and the runs with each parameter "frozen", the contribution of every parameter to GDB has been quantified. It should be noted that the sum of the partial GDB contributions of the various parameters is not exactly equal to the total GDB of the main RTM run, because the interactions between these parameters and radiation, for example, aerosol–cloud–radiation interactions, are not linear. The results of Table 2 indicate that on a global scale, TCC and middle cloud cover have contributed the most to the 2001–2009 GDB, specifically by −1.4 W m$^{-2}$ and −2 W m$^{-2}$, respectively. Next, aerosols, and more specifically AOT, have also contributed significantly, namely by −0.7 W m$^{-2}$, to the dimming observed in the first decade of the 21st century. High cloud cover has contributed equally to AOT, i.e., by −0.7 W m$^{-2}$, while low cloud cover has counteracted dimming, inducing an increase in SSR equal to 0.9 W m$^{-2}$. It should be noted that although the results of Tables 1 and 2 are similar, i.e., they identify TCC and AOT as the main contributors to the 2001–2009 GDB, they are not identical, since Table 2 identifies middle cloud cover as being an even stronger contributor. However, this should not be considered as a problem, since correlations between changes in parameters (ΔV) and ΔSSR and the contributions of these changes to GDB are not identical, but they have a different physical meaning. Table 2 also provides information for NH and SH separately. It appears that the global findings are more or less valid for NH, with the exception of AOT, which is not found to be a main contributor. However, the situation is different for SH, for which AOT is the secondary contributor (−1.7 W m$^{-2}$) to 2001–2009 GDB, while TCC is not so important (−1.1 W m$^{-2}$). It should be also noted that low cloud cover has significantly counteracted the overall dimming, by contributing an important brightening equal to 1.6 W m$^{-2}$. This low-level ISCCP cloud cover decrease, however, should be taken cautiously given the corresponding increase

and decrease in ISCCP mid- and high-level cloud cover and the ISCCP cloud overlap issues [53]. In Figure 5a,b, the global distribution of the computed contribution of changes in total cloud cover (ΔTCC) and aerosol optical thickness (ΔAOT) to the changes of downward surface solar radiation (ΔSSR) over each geographical cell and the period 2001–2009 are shown. The information of Figure 5a,b being given on the geographical cell-level, is more detailed than the results of Table 1, regarding the contribution of physical parameters to the 2001–2009 GDB. Positive and negative values in Figure 5a,b indicate global areas where total cloud cover and AOT changes during 2001–2009 have induced (positive contributions) or counteracted (negative contributions) the estimated GDB in the present study. In Figure 5a, it can be noticed that total cloud cover has significantly (mostly by more than 50%) contributed to GDB over both land and ocean areas in both hemispheres, like over the Pacific (including the Pacific ENSO dominated areas) and Indian Oceans, as well as over the North Atlantic Ocean and North America. On the other hand, AOT (Figure 5b) contributed to GDB mostly over continental areas of the globe, and much less over oceans. For example, a large contribution of AOT (percentages between 50% and 100%) is observed over Europe, the Mediterranean Sea, Arabia, the Arabian Sea, as well as over the Amazonia and Asian regions. On the contrary, bluish-colored areas in Figure 5b, mostly observed over oceans, indicate that changing AOT has counteracted the overall GDB therein.

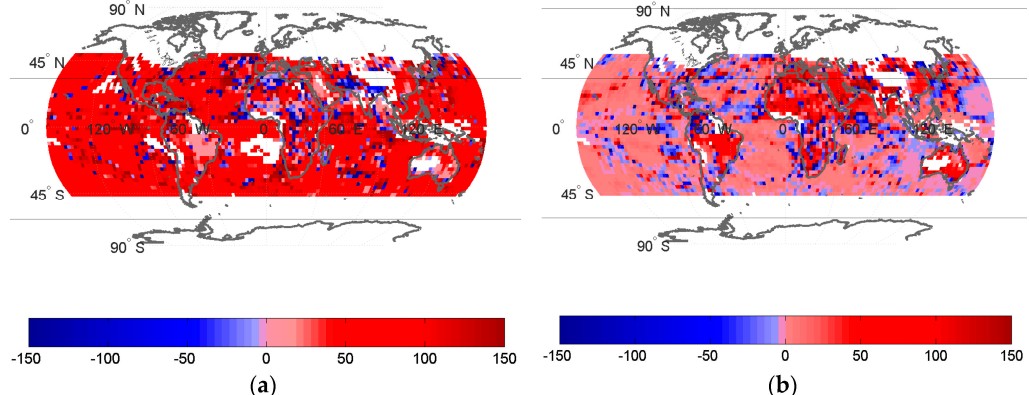

**Figure 5.** Contribution of changes in total cloud cover (**a**) and changes in aerosol optical thickness (**b**) to the model-computed changes of downward surface solar radiation (ΔSSR) over the period 2001–2009 (Figure 1). The contribution is expressed in relative percent (%) terms with respect to (ΔSSR).

**Table 1.** Correlation coefficient (R) between changes in the main physical parameters, Δ(V), and changes in downward surface solar radiation, Δ(SSR). The R values are computed for matched Δ(V) and Δ(SSR) geographical cell data pairs all over the globe. The percentages of the total number of matched data pairs lying in the 2nd and 4th quadrants (indicating physical agreement between changes of parameters and SSR) and in the 1st and 3rd quadrants (physical disagreement) are also given in the third and fourth column of the table. Colored cells indicate the three parameters having the strongest correlation with Δ(SSR). The brown color indicates the strongest, the red color indicates the second strongest, and the orange color indicates the third strongest correlation coefficients between Δ(V) and Δ(SSR).

| Comparison | R | 2nd–4th Quadrant | 1st–3rd Quadrant |
|---|---|---|---|
| Δ(SSR)-Δ(Total Cloud cover) | −0.58 | 71.40% | 28.60% |
| Δ(SSR)-Δ(High Cloud cover) | 0.02 | 48.20% | 51.80% |
| Δ(SSR)-Δ(Middle Cloud cover) | −0.30 | 69.20% | 30.80% |
| Δ(SSR)-Δ(Low Cloud cover) | −0.37 | 56.40% | 43.60% |
| Δ(SSR)-Δ(Aerosol Optical Thickness) | −0.25 | 63.20% | 36.80% |
| Δ(SSR)-Δ(Optical Thickness Absorption High Cloud) | −0.47 | 74.80% | 25.20% |
| Δ(SSR)-Δ(Optical Thickness Absorption Middle Cloud) | −0.32 | 68.00% | 32.00% |
| Δ(SSR)-Δ(Optical Thickness Absorption Low Cloud) | −0.24 | 63.30% | 36.70% |
| Δ(SSR)-Δ(Optical Thickness Scattering High Cloud) | −0.44 | 76.00% | 24.00% |
| Δ(SSR)-Δ(Optical Thickness Scattering Middle Cloud) | −0.31 | 69.00% | 31.00% |
| Δ(SSR)-Δ(Optical Thickness Scattering Low Cloud) | −0.24 | 61.60% | 38.40% |
| Δ(SSR)-Δ(Precipitable Water) | −0.28 | 61.40% | 38.60% |

**Table 2.** Contribution (in W/m$^2$) of changes in the main drivers of surface solar radiation to the changes of downward surface solar radiation (DSR), i.e., GDB, during 2001–2009; the results are given separately for the two hemispheres and the globe (for the areal coverage of Figure 1). Colored cells indicate the largest contributions to GDB. Brown, red, and orange colors indicate the largest, second largest, and third largest contributions to dimming (negative values), respectively, and the cyan color corresponds to brightening (positive values).

| Causes | Global | NH | SH |
|---|---|---|---|
| CONTRIBUTION OF TOTAL CLOUD COVER TO GDB | −1.4 W/m$^2$ | −1.4 W/m$^2$ | −1.1 W/m$^2$ |
| CONTRIBUTION OF HIGH CLOUD COVER TO GDB | −0.7 W/m$^2$ | −0.6 W/m$^2$ | −0.8 W/m$^2$ |
| CONTRIBUTION OF MIDDLE CLOUD COVER TO GDB | −2 W/m$^2$ | −1.7 W/m$^2$ | −2.4 W/m$^2$ |
| CONTRIBUTION OF LOW CLOUD COVER TO GDB | 0.9 W/m$^2$ | 0.7 W/m$^2$ | 1.6 W/m$^2$ |
| CONTRIBUTION OF HIGH CLOUD OPTICAL THICKNESS TO GDB | −0.5 W/m$^2$ | −0.4 W/m$^2$ | −0.5 W/m$^2$ |
| CONTRIBUTION OF MIDDLE CLOUD OPTICAL THICKNESS TO GDB | −0.1 W/m$^2$ | −0.05 W/m$^2$ | −0.1 W/m$^2$ |
| CONTRIBUTION OF LOW CLOUD OPTICAL THICKNESS TO GDB | 0 W/m$^2$ | −0.03 W/m$^2$ | 0.02 W/m$^2$ |
| CONTRIBUTION OF AEROSOL OPTICAL THICKNESS TO GDB | −0.7 W/m$^2$ | −0.2 W/m$^2$ | −1.7 W/m$^2$ |
| CONTRIBUTION OF PRECIPITABLE WATER TO GDB | −0.1 W/m$^2$ | −0.13 W/m$^2$ | 0.04 W/m$^2$ |

## 4. Conclusions

According to recent studies, a global brightening in the 1990s was preceded by a global dimming from the middle of the 20th century through the 1980s. Some studies using data beyond 2000 examined the tendencies of SSR up to 2004 on the global/hemispherical/continental scales or station sites, indicating a new change in GDB from the 20th to the 21st century. Given the decadal scale of GDB, its possible impact on global warming [54], and the recent global warming hiatus [55] in the first decade of 21st century, we further analysed SSR over various world regions in the present study in order to investigate more reliably the SSR patterns beyond 2000 and through the end of the 2000s all over the globe. Special care was taken in order to ensure reliable tendencies of SSR, by applying severe criteria on the completeness of the time series of SSR anomalies, by examining the statistical significance of the computed tendencies and by comparing against corresponding time series from the reference GEBA and BSRN surface station networks.

A statistically significant overall global dimming is found to have taken place on Earth under all-sky conditions from 2001 to 2009 arising from a stronger solar dimming in the SH ($\Delta$SSR = −3.1 W m$^{-2}$ or −0.34 W m$^{-2}$/year) and a weaker dimming in NH ($\Delta$SSR = −2.2 W m$^{-2}$ or −0.24 W m$^{-2}$/year), which are also statistically significant. Dimming is found to have taken place over both land and ocean areas in both hemispheres, though it is more distinct over SH continental areas ($\Delta$SSR = −14.2 W m$^{-2}$ or −1.58 W m$^{-2}$/year). The regional patterns of $\Delta$SSR, however, have a patchy spatial structure, with opposite SSR tendencies over neighbouring areas, even on a continental scale. Thus, although SSR has increased over the eastern Atlantic coastal, and also over the USA coastal areas around the Gulf of Mexico, SSR has decreased over the western and central (middle) USA areas. Mixed SSR patterns also appear over South America, Africa, and Asia, although dimming predominates over all of them. On the other hand, there are some areas characterized by clear brightening, such as Europe and the Mediterranean. The model-computed tendencies of SSR, which are found to be statistically significant in 26% of the total number of cells with computed results, namely over southern Europe, the Arabian Peninsula, China, Australia, South America, and broad areas of the Pacific and South Atlantic Oceans, are supported to a large degree by surface station measurements taken from the GEBA and BSRN networks, which strengthens the validity of the post-2000 GDB findings of this study.

Clouds appear to have been primarily responsible for GDB in the first decade of the 21st century, with aerosols playing a secondary role on a hemispherical/global basis. More specifically, middle cloud cover is found to have the single largest contribution to the post-2000 GDB (−2.0 W m$^{-2}$ on global scale, and −1.7 and −2.4 W m$^{-2}$ in NH and SH, respectively). Aerosols, namely AOT, also had an important contribution, equal to −0.7 W m$^{-2}$, much smaller in NH than SH (−0.2 and −1.7 W m$^{-2}$, respectively) and much larger over land than ocean areas. On the contrary, low cloud cover is found to have counteracted the overall dimming, having caused an average global increase in SSR equal

to 0.9 W m$^{-2}$, thus underlining the complex role and radiative effects of clouds, depending on their specific type.

The identified dimming during the first decade of the 21st century in this study marks a new change in decadal scale GDB patterns, following the brightening observed during the 1990s and a prior dimming up to the 1980s, which coincides with a slowdown of the increase rate of the otherwise relentless global warming during the 2000s. Such inter-decadal patterns of GDB are important and need to be continuously monitored in order to gain more insight into their nature, causes, and climatic implications.

**Supplementary Materials:** The following are available online at http://www.mdpi.com/2073-4433/11/3/308/s1, Figure S1: Comparison between the variability of time series of model-computed and station deseasonalized anomalies of monthly SSR fluxes during 2001–2009 as quantified with the difference between their 95% Confidence Intervals (95% CI). The size of circles indicates the magnitude of computed percent differences between model and station 95% CI (confidence interval) values over the globe for 20 BSRN (**a**) and 105 GEBA (**b**) stations. Blue and red circles indicate BSRN and GEBA stations for which model computations and station measurements agree and disagree, respectively, Figure S2: Time series of deseasonalized anomalies of monthly SSR fluxes averaged over land (green lines), ocean (blue lines) and land + ocean (black lines) regions of the northern hemisphere (**a**), and southern hemisphere (**b**), over the period 2001–2006. The GDB magnitudes (ΔSSR, in W m$^{-2}$) and the associated standard error values) over the period 2001–2006 are also given for each hemisphere.

**Author Contributions:** Conceptualization, N.H.; methodology, N.H.; software, N.H., C.D.P., I.V. and C.M.; validation, E.I., C.M., M.-B.K.-C. and M.G.; formal analysis, N.H. and E.I.; investigation, E.I. and N.H.; resources, N.H., C.M., M.W. and I.V.; data curation, E.I., M.-B.K.-C., M.G. and M.W.; writing—original draft preparation, E.I.; writing—review and editing, N.H., A.F., C.D.P., C.M., N.B., M.W. and I.V.; visualization, E.I., M.-B.K.-C.; supervision, N.H.; project administration, N.H.; funding acquisition, N.H. All authors have read and agreed to the published version of the manuscript.

**Funding:** This research received no external funding.

**Acknowledgments:** The ISCCP D2 and Collection 006 MODIS Level-3 mean monthly AOD data were obtained from the NASA's ISCCP web site http://isccp.giss.nasa.gov and MODIS Data Processing System (MODAPS), respectively. NCEP/NCAR Reanalysis Derived data were provided by the NOAA/OAR/ESRL PSD, Boulder, Colorado, USA (http://www.cdc.noaa.gov/). The Global Energy Balance Archive (GEBA) were taken from the GEBA website maintained at ETH, Switzerland (http://www.geba.ethz.ch/) and the Baseline Surface Radiation Network (BSRN) data were taken from the World Radiation Monitoring Center (WRMC), central archive of the Baseline Surface Radiation Network (BSRN) PANGAEA ftp site (https://bsrn.awi.de/data/data-retrieval-via-pangaea/). GEBA is co-funded by the Federal Office of Meteorology and Climatology MeteoSwiss within the framework of GCOS Switzerland.

**Conflicts of Interest:** The authors declare no conflict of interest.

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
