# Peer review of "Global Dimming and Brightening Features during the First Decade of the 21st Century"

_atmosphere, doi:10.3390/atmos11030308_

Round 1
Reviewer 1 Report
This manuscript deals with the brightening and dimming (SSR trends) in the first decade of 21st century. The topic is very interesting, and yet I have some reservations. I think the work has potential but the manuscript needs to be rewritten. I'll list now things that need changing.
1. Material and methods needs to be much more thorough. There has to be an explanation of the instruments used to acquire the data in the datasets mentioned, and more detailed explanation of the RTM and the parameters used in it (no matter that they have been explained in another paper) is needed. Also, this section should include a thorough description of the analysis performed to achieve the results (how are anomalies calculated? how are trends calculated?). This is somehow mentioned in the Results section, but it should be detailed in this section instead of the results section.
2. English must be improved.
3. Maybe "not shown" results can be added to "Supplementary material"?
4. I am missing a discussion of how representative is this study of the whole planet. For instance, are the means presented representative of the NH and SH? The study is lacking, after all, polar regions.
5. L. 295-297. Authors claim that other atmospheric parameters, like cloud optical thickness and precipitable water should be condiered...why are they not considering them?
Author Response
We would like to thank Reviewer 1 for his/her comments and suggestions. We have tried to address the raised issues. Please see below out point-by-point answers (normal font) to the Reviewer’s comments (italic font). Please, note that the reported line numbers below refer to the tracked changes formatted revised paper.
- Material and methods needs to be much more thorough. There has to be an explanation of the instruments used to acquire the data in the datasets mentioned, and more detailed explanation of the RTM and the parameters used in it (no matter that they have been explained in another paper) is needed. Also, this section should include a thorough description of the analysis performed to achieve the results (how are anomalies calculated? how are trends calculated?). This is somehow mentioned in the Results section, but it should be detailed in this section instead of the results section.
The Materials and Methods section (number 2) has been enriched, as suggested by the Reviewer. More specifically:
- a more detailed description of the utilized RTM and its input data is provided (lines 84-123).
- a description of the BSRN and GEBA station networks along with the adopted methodology of taken measurements and subsequent processing by them is given in section 2 (lines 130-150).
- reference is now made (lines 155-174) to the applied analysis, explaining how the surface solar radiation (SSR) anomalies are computed and providing a brief description of the applied linear regression and trend model for the computation of SSR tendencies. Reference is also made to the applied non parametric Mann-Kendall test for the examination of the statistical significance of the computed SSR tendencies.
- English must be improved.
The use of English language has been improved and relevant corrections were made in the text.
- Maybe "not shown" results can be added to "Supplementary material"?
In order to respond to this request, a Supplement has been prepared and complements now the manuscript. In the Supplement are provided:
- results (Figure S1) confirming the ability of the RTM to match the variability of the GEBA and BSRN station SSR time series. More specifically, in Figure S1 is compared the variability of model-computed and station SSR time series during 2001-2009 as quantified with the difference between their 95% Confidence Intervals (95% CI).
- results that support the investigation of the reasons (different time periods and geographical coverage) for the encountered differences between the SSR tendencies computed in the present study (for the period 2001-2009) and in our 2012 study (for the period 2001-2006). More specifically, in Figure S2 are given the time series of deseasonalized anomalies of monthly SSR fluxes averaged over land, ocean and land + ocean regions of the northern and southern hemispheres, over the period 2001-2006.
As far as it concerns the scatter plots between ΔV, i.e. the changes of every parameter, and ΔSSR, i.e. the changes of SSR, we believe that showing this large number of scatterplots would not add any significant additional information to that provided by Table 1, which effectively summarizes their statistical metrics. We remind that Table 1 lists the computed correlation coefficients (R), and the agreement (percentage of total number of matched data pairs lying in the 2nd and 4th quadrants) and disagreement (percentage of points in the 1st and 3rd quadrants) between the tendencies ΔV of the most significant parameters and SSR.
- I am missing a discussion of how representative is this study of the whole planet. For instance, are the means presented representative of the NH and SH? The study is lacking, after all, polar regions.
Indeed, the present study lacks polar regions. We would like to remind that this is due to the application of the severe criterion of availability of 100% of all RTM input data and computed SSR fluxes in order to ensure reliable GDB results (statistical significance and robustness of computed SSR tendencies). We just preferred to produce more reliable results, even if they are spatially limited lacking polar areas, than obtaining results over these latitudes that would be unreliable.
We acknowledge that given this spatial limitation the results are not representative of the entire hemispheres and the globe. However, we believe that the study fairly provides reliable GDB results over the covered world regions. The maps allow the assessment of GDB over specific locations (geographical cells), while the so-called hemispherical results are clearly labeled as referring to specific parts of the hemispheres (0°-50°N and 0°-45°S).
- L. 295-297. Authors claim that other atmospheric parameters, like cloud optical thickness and precipitable water should be condiered...why are they not considering them?
Actually, these parameters (and others) are considered for explaining the GDB patterns. It is exactly what is made in the following discussion, after the part of the manuscript mentioned by the Reviewer (lines 295-297 of original manuscript). More specifically, in lines 390-443 of the manuscript, the discussion (based on the results of Tables 1 and 2) focuses on the assessment of the role of different parameters for GDB
Probably, the wording in the previous version of the manuscript was misleading and thus we tried to modify it in order to avoid misunderstanding. Thus, the specific part of the manuscript (lines 383-385) now reads as: “Hence, other atmospheric parameters, like cloud optical thickness and precipitable water, should bewere considered in order to explain the GDB patterns.”

Reviewer 2 Report
Please find my comments and suggestions within the attached pdf file.

Author Response
We would like to thank Reviewer 2 for his/her comments and suggestions, which were taken into account in the revised manuscript.
Detailed point-by-point answers to the Reviewer comments, which were made in his/her pdf file, are provided in the annotated pdf file (the answers to the comments are posted within it). Therefore, only a very brief note is made here (below) referring to the modifications made in the revised manuscript (with the reported line numbers referring to the tracked changes formatted revised paper).
We tried to address all the raised issues and comments made. For example, reference is made to the presence/absence of statistical significance in the computed SSR tendencies by the RTM, at the beginning of the discussion of GDB patterns (first paragraph of section 3.1, lines 177-180 and 189-198). Also, reference is made to the use of the Deep Blue Algorithm in MODIS C006 data, citing the works of Hsu et al. or Sayer et al., in section 2 (lines 102-103). In addition, reference is made to the study from Pozzer et al, 2015, ACP for explaining the computed dimming over the Sahara and Arabian peninsula (lines 244-246).

Round 2
Reviewer 1 Report
The paper has improved a lot, still I would make some minor changes to get it ready for publication:
- I have some concerns about using monthly data.
- Materials and methods would be much easier to read if divided in parts (Data, RTM model, statistics). I miss some information still, like how are hemispheres averaged, how are correlations coefficients computed (a single linear model? a multivariate model?).
- Maybe the authors should consider changing their "linear model trend estimate" to a non-parametric model, like Sen's slope, or alternatively check if their data are normally distributed.
- Improve quality of Figures
- I would consider give the "validation" part a subsection in the results section. The paper would be much clearer in my opinion.
Minor changes
- Abstract: specify if Wm^2 are per year or in which period.
- L34: add a comma afther "(SSR)" : "The downward surface solar radiation (SSR)**,** apart from long-term variations, has been shown"
- L41: there is a change of font in this line
- L46: authors could specify which regions
- L56-60: maybe change "through to" to "until" or "up to"
- L83 "is the same as"
- Tables are not properly formated with their captions, etc.
